# A Comparison of the Effect of Two Types of Whole Body Vibration Platforms on Fibromyalgia. A Randomized Controlled Trial

**DOI:** 10.3390/ijerph18063007

**Published:** 2021-03-15

**Authors:** José Antonio Mingorance, Pedro Montoya, José García Vivas Miranda, Inmaculada Riquelme

**Affiliations:** 1Research Institute of Health Sciences (IUNICS-IdISBa), University of the Balearic Islands, 07122 Palma de Mallorca, Spain; pedro.montoya@uib.es (P.M.); inma.riquelme@uib.es (I.R.); 2Department of Nursing and Physiotherapy, University of the Balearic Islands, 07122 Palma de Mallorca, Spain; 3Laboratory of Biosystems, Institute of Physics, Federal University of Bahia, Salvador 40170-115, Brazil; vivasm@gmail.com

**Keywords:** chronic pain, proprioception, quality of life, postural balance

## Abstract

Whole body vibration has been proven to improve the health status of patients with fibromyalgia, providing an activation of the neuromuscular spindles, which are responsible for muscle contraction. The present study aimed to compare the effectiveness of two types of whole body vibrating platforms (vertical and rotational) during a 12-week training program. Sixty fibromyalgia patients (90% were women) were randomly assigned to one of the following groups: group A (*n* = 20), who performed the vibration training with a vertical platform; group B (*n* = 20), who did rotational platform training; or a control group C (*n* = 20), who did not do any training. Sensitivity measures (pressure pain and vibration thresholds), quality of life (Quality of Life Index), motor function tasks (Berg Scale, six-minute walking test, isometric back muscle strength), and static and dynamic balance (Romberg test and gait analysis) were assessed before, immediately after, and three months after the therapy program. Although both types of vibration appeared to have beneficial effects with respect to the control group, the training was more effective with the rotational than with vertical platform in some parameters, such as vibration thresholds (*p* < 0.001), motor function tasks (*p* < 0.001), mediolateral sway (*p* < 0.001), and gait speed (*p* < 0.05). Nevertheless, improvements disappeared in the follow-up in both types of vibration. Our study points out greater benefits with the use of rotational rather than vertical whole body vibration. The use of the rotational modality is recommended in the standard therapy program for patients with fibromyalgia. Due to the fact that the positive effects of both types of vibration disappeared during the follow-up, continuous or intermittent use is recommended.

## 1. Introduction

Fibromyalgia (FM) is a chronic syndrome characterized by widespread pain sensitivity, fatigue, and cognitive and affective symptoms, affecting women predominantly [1]. The prevalence of fibromyalgia in Spain is 2.4% (95% CI: 1.5–3.2). The Balearic Islands have about 20,000 patients with fibromyalgia. This disease is more frequent in women (4.2%) than in men (0.2%), which implies a female/male 21:1 ratio [2]. Other symptoms such as sleep disturbances, morning stiffness, and psychological disorders such as anxiety and depression are often associated [2,3]. Fibromyalgia affects 3–5% of general population, occurring in all ages, with chronic symptoms that can fluctuate throughout the day, and with inactivity compromising about a quarter of those affected [4].

Whole body vibration (WBV) training is defined as the use of a stimulus provided by a vertical or rotating oscillation platform while the individual attempts to maintain a position. The individual stands on the platform and the oscillations cause vibrations that are transmitted to the subject through the legs. WBV has been used in different fields, from the training of elite athletes to the treatment of patients with chronic pain [5]. Regarding the use of WBV for the management of chronic pain, studies in osteoarthritis or rheumatoid arthritis reported improvements in pain, strength, and functionality [6,7,8,9]. These studies attributed these improvements to the “tonic vibratory reflex”, which increases the recruitment of motor units and induces a more efficient use of the positive proprioceptive feedback loop. Likewise, through this therapy, improvements in chronic low back pain have also been observed, in terms of pain and disability [10,11].

Positive effects of WBV have been observed on the neuromuscular system. Mechanical vibrations applied to both the muscle and the tendon cause the tonic vibratory reflex, with an activation of the muscle spindles and therefore an improvement of the stretch reflex. This vibration-generated stimulation in the musculature through the tonic vibration reflex produces an inhibitory effect on neurons of the spinothalamic tract, leading to a reduction in pain intensity [12].

Given that WBV represents an effective nonpharmacologic intervention, there is a need for more critical evaluation. Two types of platforms can deliver WBV. One is a rotational vibration device that induces reciprocal displacements on the left and right sides of a fulcrum and generates higher lateral than vertical acceleration [13]. The second one is a vertical vibration device that induces up-and-down oscillations (approximately 1 to 2 mm) over a vertical axis and produces greater strain in the vertical than in the lateral axis [14].

The transmission of vibration to the human body is a complicated phenomenon due to nonlinearities in the musculoskeletal system and it depends on the intensity and direction of the vibration [15]. Besides, the transmission of rotational WBV is also influenced by body posture [16]. It has been observed that the acceleration transmitted to the different parts of the body is greater using rotational WBV than vertical WBV during the standing position, so a lower frequency and a higher amplitude in rotational WBV in the standing position is recommended to establish a desired level of mechanical load in different areas of the body [17]. Conversely, the vertical WBV induces very stable vibration patterns, moving completely in the vertical direction and showing very few horizontal vibrations. This is perceived by the patient as more comfortable and less challenging, because the platform acts in the direction of gravity and does not stress the body with unnatural forces. Furthermore, because it does not use large amplitudes of vibration (1–2 mm), the acceleration it can generate is limited [18,19].

Although the use of whole body vibration is recommended in patients with fibromyalgia, there is very little evidence and studies are of limited quality, with few participants, wide confidence intervals, lack of generalizability (all studies were in women and not in men), and some aspects related to the risk of bias [20], like, for example, blinding of participants and personnel [21,22], blinding of self-reported outcome [21], or selective reporting [23]. Furthermore, these studies did not measure important parameters such as motor function, proprioception, or vibration-sensitive threshold. Some reviews [20,24,25] have highlighted the need for further research in this field to improve understanding of the effects of WBV in patients with fibromyalgia and have suggested that additional research needs to include different devices with the same protocol, as well as among different groups of patients in terms of gender. In our study, 10% of participants were men, according to the gender-related epidemiology of fibromyalgia [1].

There are no studies of patients with fibromyalgia with different devices in the same protocol, so the differences between both types of vibration, in terms of the effectiveness of treatment in patients with fibromyalgia, have not been defined. Given this context, the objective of this randomized controlled trial was to compare the effects of two types of WBV on patients with fibromyalgia through a 12-week therapy program, compared with a control group that did not perform the therapy program. Effects were examined in a wide range of symptoms, such as health-related quality of life, pain intensity, somatosensory sensitivity, motor function, muscle strength, and balance.

## 2. Materials and Methods

### 2.1. Participants

A single blind randomized controlled trial was performed The maximum possible variability (variance) was assumed, that is, *p* = 0.50 and a maximum sampling error of 0.08, while the number of patients to be evaluated was calculated by adopting a variance level of 0.95%. Considering that there are 20,000 patients with fibromyalgia in the Balearic Islands [2], it was determined that 60 patients were required to obtain a significant sample. Equation (1) was used to estimate the sample size n.
(1)$$n=\frac{N\;Z^{2}\;p\;q}{d^{2}\;\left(N-1\right)+\;Z^{2}\;p\;q},$$
where *N* = population size, *Z* = confidence coefficient for a given confidence level, p = probability of success, q = probability of failure, and *d* = maximum permissible error.

Consequently, sixty patients with fibromyalgia, diagnosed by physicians according to the American College of Rheumatology 2016 criteria [1], were recruited from different fibromyalgia associations in Mallorca by an external researcher. At the time of recruitment, all participants were verbally informed about the details of the study and provided written consent. The study was approved by the Ethics Committee of the Balearic Islands (Spain) (reference IB-2586/15 PI) and was registered as a clinical trial ID with reference number NCT03782181.

Randomization was performed in two stages: generation of numbers (table of random numbers) and blind allocation (opaque and sealed envelopes). The envelopes indicating which group (rotational vibration, vertical vibration, or control) the participants would be included in were opened after written informed consent was obtained from each participant. The outcome assessors were not informed about the allocation of patients in the respective groups.

The sixty participants met the inclusion criteria: (1) age between 30 and 65 years, and (2) diagnosis of fibromyalgia according with the American College of Rheumatology 2016 criteria for fibromyalgia [1]. Exclusion criteria were: history of severe trauma, peripheral nerve entrapment, inflammatory rheumatic diseases, severe disease that prevents supporting the program’s physical load, pregnancy, participation in a psychological or physical therapy program, or participation in regular physical exercise more than once a week over a 2-week period in the last 5 years.

Participants were randomly allocated into one of three groups: a group (*n* = 20) that experimented with vertical WBV, a group (*n* = 20) that experimented with rotational WBV, and a control group (*n* = 20) that did not perform any training. The three groups were matched for age, weight, height, body mass index, and pain duration (Figure 1).

Table 1 displays the characteristics of the participants of the three groups. Regarding medication intake, most patients were taking pain medication such as analgesics, anxiolytics, and antidepressants. For medical and ethical reasons, medication was not discontinued during the study.

### 2.2. Instruments and Procedure

#### Intervention Procedure

The intervention groups performed a training program in accordance with the parameters established by previous literature on vertical or rotational WBV [26]. In our study, we used the Galileo model (Novotec Medical GmbH, Pforzheim, Germany) for rotational WBV, which is used as exercise equipment and for therapeutic use. It consists of a vibration platform that vibrates sinusoidal side, alternating like a see-saw. This platform generates vibration by rotating along the sagittal axis and the frequency is adjustable within a range of 5–30 Hz [27].

For vertical WBV, the Power Plate model (Power-Plate International B.V., Badhoevendorp, The Netherlands) was used, which moves synchronously in vertical direction. Its frequency is adjustable within a range of 25–50 Hz [28]. For a direct comparison of the two types of vibration, the two devices were calibrated, setting the frequency to 25 Hz and the foot-to-foot distance was set to 21 cm. Each session consisted of three sets of 45 s with 120 s recovery between sets. It was a standardized program performed at the same time in the afternoon and it consisted of maintaining three different static positions on the platform during vibration: standing with both feet on the platform and unilateral static squat position, 22 s with the right leg and 22 s on the left leg. The therapeutic program was performed in an air conditioned room by an external physiotherapist trained for all conditions and lasted for 12 weeks, with a frequency of three sessions per week. Participants in the control group did not perform any WBV program. Assessment of outcomes of the three groups was undertaken by an external researcher at baseline (T1, pretreatment), immediately after the therapy program (T2, post-treatment), and three months after the end of the therapy (T3, follow-up). None of the volunteers underwent physiotherapy treatment or physical exercise before the intervention. The level of previous functionality was verified by functional tests such as the Berg Scale, the six-minute walking test, and the isometric strength of the back muscles.

### 2.3. Self-Report Questionnaires

Fibromyalgia Impact Questionnaire [29]. This is a commonly used instrument in the evaluation of patients with fibromyalgia over 10 dimensions (i.e., functional capacity, feeling good, work absenteeism, interference of symptoms at work, pain, fatigue, morning stiffness, morning tiredness, anxiety, and depression). A higher score indicates a greater impact on the person. Its reliability is >0.80 and its validity is 0.93 [29].

Visual Analogue Pain Scale [30]. Each participant was asked to indicate their current level of pain using a 10-cm line from 0 (no pain) to 10 (highest level of pain). This has been reported to be a reliable and valuable method for assessing pain. The reliability values range from 0.60 to 0.77 and its validity is 0.51 [30].

Quality of Life Index [31]. It is composed of 10 dimensions, including aspects ranging from physical well-being to spiritual fulfillment, as well as a global perception of quality of life. It has been used to assess quality of life in patients with fibromyalgia [32]. A higher score is indicative of a higher quality of life. Its reliability correlation coefficient is 0.89 and its validity is 0.90 [31].

### 2.4. Sensitivity Measures

Pressure pain sensitivity was assessed by means of three trials per location (expressed in Newtons) through an electronic algometer (Force One, Wagner Instruments, Riverside, CA, USA) [33]. The patient was instructed to say stop as soon as the perception of the stimulus changed from pressure to pain. Pressure stimuli were applied on two bilateral body locations: epicondyles (related to the pain area in fibromyalgia) and index fingers (not related to the pain area). Reliability values range from 0.990 to 0.999 and its validity is 0.323 [33].

Vibration thresholds at the big toes and at the index fingers were quantified bilaterally using a Vibratron II device (Physitemp, Clifton, NJ, USA). Using a two-alternative forced-choice procedure, subjects identified which of two rods was vibrating. The control unit displayed scores in vibration units (the amplitude of vibration, proportional to the square of applied voltage) [34]. The testing started with an intensity above the threshold, and then it was gradually reduced, asking participants to indicate when the vibration was not perceived. Its reliability is >0.7 and its validity is 0.93 [34].

### 2.5. Motor Function Tasks

Berg scale. This is a functional assessment tool, consisting of scores ranging from 0 (impaired) to 56 (excellent, highest level of function). It has been used in patients with fibromyalgia to assess balance [35]. Its reliability is high, with a pooled estimate of 0.97 (95% CI 0.96 to 0.98) and its validity is 0.953 (95% CI 0.910 to 0.975).

Six-minute walking test. Its purpose is to measure the maximum distance that a patient can walk over a period of six minutes walking as fast as possible. The test has been validated in patients with fibromyalgia [36,37]. Its reliability ranges from 0.91 to 0.98 and validity is 0.657 [36].

Isometric back muscle strength was determined as the maximal isometric strength of the trunk muscles in standing posture with 30° lumbar flexion using a dynamometer (T.K.K.5002). It has been used in patients with fibromyalgia to assess back muscle strength [38]. Its reliability is 0.930 (95% CI 0.805 to 0.975) and its validity is 0.92 [38].

### 2.6. Static and Dynamic Balance

Static balance was assessed by using a modified version of the Romberg’s balance test [39]. Central postural control is dependent on input from three peripheral modalities: vision, vestibular apparatus, and proprioception. Disturbance in any one of these modalities can be compensated by inputs from the other two systems, thus, impaired postural balance can be overcome by visual and vestibular feedback. Asking the participants to close their eyes during the Romberg’s test helps uncover any disordered proprioception that may have been masked by vision, so the patient should become more unsteady with eyes closed. In the present study, we analyzed the oscillatory body movements during the test performance. Participants were situated below a standard webcam (© Logitech) situated above the ground and placed at a mean distance of 50 cm from the participant’s head. The participant was asked to remain in orthostatic position with feet parallel at shoulder height, arms extended along the body and eyes closed for 1 min [40]. Motion on the frontal and sagittal planes was captured by the webcam at 30 frames per second. For analyses of motion parameters, a free open source software for computer vision analysis of human movement was used (CvMob 3.1). This software has a high degree of accuracy for calculating body position and movement in the X and Y axes and produces similar results to posturography in the analyses of body balance [41]. Velocity and body sway in the anteroposterior and mediolateral directions were extracted. Its interclass correlation coefficient is 0.87 and its validity is 0.87 [39].

Dynamic balance was assessed by means of a gait task. Participants were instructed to walk on a 4 m carpet at their normal walking step, with shocks and with flexed arms positioned on the abdomen. Optical markers were attached at the following three body positions: area between the lateral condyle of the femur and the fibular head, great trochanter, and lateral malleolus. Each subject’s motion was digitally recorded with a video camera at 210 frames per second (CasioExilimEX-FS10). The camera was positioned at a distance of 4 m from the carpet to visualize changes in position, velocity, and anatomical points along the *x*-axis. An open-source software (CvMob 3.1) for computer vision analysis of human movement [41] was used to extract the following variables: gait velocity (cm/s), stride length (cm), and percentages of time in the stance/swing phase. The CvMob is a reliable tool for analysis of linear motion and lengths in two-dimensional evaluations of human gait. A strong correlation (rs mean = 0.988) of the linear trajectories between systems and inter- and intrarater analysis were found and its validity is 0.85 (95% CI 0.80 to 0.90) [41].

Analyses were performed using two-way analysis of variance (ANOVA), with the between-factor GROUP (intervention vertical vs. intervention rotational vs. control) and the within-subject factor TIME (pretreatment vs. post-treatment vs. follow-up). The significance level was set at 0.05. A 5% margin of error and 95% confidence interval were considered. All the analyses were performed using SPSS Statistics 20. We checked the normal distribution of the residuals of all variables and found that this assumption was fulfilled in following variables: isometric back muscle strength, Fibromyalgia Impact Questionnaire, pressure pain sensitivity, vibration thresholds, and stride length. For the rest of the variables in which the Shapiro–Wilk test revealed a violation of the normality assumption, the histogram and the normal probability plot was checked and in most of the cases only one or two outliers were observed.

## 3. Results

The sixty subjects enrolled in the groups adhered totally to the program, with no occurrence of sample loss: none of the subjects dropped out of the study in any of the time periods. The three groups were similar in their sociodemographic characteristics (all *p* > 0.05, Table 1). The mean age was 52.4 ± 8.4 years, ranging from 35–65 years; most of the participants were female (90%). The mean duration of pain was 7.3 years, with an average of three years from the clinical diagnosis of fibromyalgia. Table 2 displays the descriptive data of the three groups in the three assessment times.

### 3.1. Self-Report Questionnaires

Regarding the Fibromyalgia Impact Questionnaire scores, the ANOVA revealed significant effects due to TIME (2, 114) = 54.16, *p* < 0.001) and GROUP x TIME (F(4, 114) = 28.43, *p* < 0.001). Post hoc comparisons revealed that participants in both intervention groups significantly decreased their scores from the pre- to the post-treatment sessions (*p* < 0.05), but these scores increased from the post-treatment to the follow-up (*p* < 0.05). No significant change was observed in the control group. There were no significant effects due to GROUP (*p* > 0.05).

For the Visual Analogue Pain Scale, the ANOVA revealed significant effects due to TIME (2, 114) = 24.74, *p* < 0.001) and GROUP x TIME (F(4, 114) = 9.67, *p* < 0.001). Post hoc comparisons revealed that participants in both intervention groups significantly decreased their scores from the pre- to the post-treatment sessions (*p* < 0.05), but these scores increased from the post-treatment to the follow-up (*p* < 0.05). There were no significant effects due to GROUP and no significant change was observed in the control group (*p* > 0.05).

For the Quality of Life Index, the ANOVA revealed significant effects due to TIME (2, 114) = 36.09, *p* < 0.001) and GROUP x TIME (F(4, 114) = 16.90, *p* < 0.001). Post hoc comparisons revealed that participants in both intervention groups significantly increased their scores from the pre- to the post-treatment sessions (*p* < 0.05), but these scores decreased from the post-treatment to the follow-up (*p* < 0.05). There were no significant effects due to GROUP and no significant change was observed in the control group (ps > 0.05).

### 3.2. Sensitivity Measures

For pressure pain thresholds, the ANOVA revealed significant effects due to TIME (Fs (2, 114) > 64.84, ps < 0.001) and GROUP x TIME (Fs(4, 114) > 38.64, ps < 0.001) in both body locations. Post hoc comparisons revealed that participants in both intervention groups significantly increased their scores from the pre- to the post-treatment sessions (*p* < 0.05), but these scores decreased from the post-treatment to the follow-up (*p* < 0.05). No significant change was observed in the control group and no significant effects were due to the GROUP (ps > 0.05).

For vibration thresholds, the ANOVA showed significant effects based on TIME in toes and index fingers (both F(2, 114) > 24.6, both *p* < 0.001) and GROUP x TIME in both locations (Fs(4, 114) > 14.8, ps < 0.001). Post hoc comparisons revealed that participants in the rotational WBV group significantly decreased their scores from the pre- to the post-treatment sessions (*p* < 0.001), but these scores increased from the post-treatment to the follow-up (*p* < 0.001). No significant change was observed in the vertical WBV or control groups (*p* > 0.05). There were no significant effects due to GROUP.

### 3.3. Motor Function Tasks

For the Berg Scale, the ANOVA revealed significant effects due to TIME (2, 114) = 128.1, *p* < 0.001) and GROUP x TIME (F(4, 114) = 85.08, *p* < 0.001). Post hoc comparisons revealed that only the participants in the rotational WBV group significantly increased their scores from the pre- to the post-treatment sessions (*p* < 0.001), but these scores decreased from the post-treatment to the follow-up (*p* < 0.001). No significant change was observed in the vertical WBV or control groups (*p* > 0.05). There were no significant effects due to GROUP.

For the six-minute walking test, the ANOVA revealed significant effects due to TIME (2, 114) = 36.57, *p* < 0.001) and GROUP x TIME (F(4, 114) = 20.14, *p* < 0.001). Post hoc comparisons revealed that only the participants in the rotational WBV group significantly increased their scores from the pre- to the post-treatment sessions (*p* < 0.001), but these scores decreased from the post-treatment to the follow-up (*p* < 0.001). No significant changes were observed in the vertical WBV or control groups (*p* > 0.05). No significant effects were due to the GROUP.

For the isometric back muscle strength, the ANOVA revealed significant effects due to TIME (2, 114) = 19.64, *p* < 0.001) and GROUP x TIME (F(4, 114) = 9.79, *p* < 0.001). Post hoc comparisons revealed that only the participants in the rotational WBV group significantly increased their scores from the pre- to the post-treatment sessions (*p* < 0.001), but these scores decreased from the post-treatment to the follow-up (*p* < 0.001). No significant changes were observed in the vertical WBV or control groups and there were no significant effects due to GROUP (ps > 0.05).

### 3.4. Static and Dynamic Balance

In static balance, for the mean sway velocity, the ANOVA revealed significant effects due to TIME (2, 114) = 12.06, *p* < 0.001) and GROUP x TIME (F(4, 114) = 2.75, *p* < 0.05). Post hoc comparisons revealed that participants in both intervention groups significantly increased their scores from the pre- to the post-treatment sessions (*p* < 0.05), but these scores decreased from the post-treatment to the follow-up (*p* < 0.05). No significant change was observed in the control group and no significant effects were due to GROUP (ps > 0.05).

For the mediolateral sway, the ANOVA revealed significant effects due to TIME (2, 114) = 10.31, *p* < 0.001) and GROUP x TIME (F(4, 114) = 3.25, *p* < 0.05). Post hoc comparisons revealed that only the participants in the rotational WBV group significantly decreased their scores from the pre- to the post-treatment sessions (*p* < 0.001), but these scores increased from the post-treatment to the follow-up (*p* < 0.001). No significant change was observed in the control group and no significant effects were due to GROUP (*p* > 0.05).

Sway in the anteroposterior axis showed no significant effects based on TIME (F (2, 114) = 1.49, *p* > 0.05) or GROUP (*p* > 0.05), nor was there an interaction effect for GROUP x TIME (F (4, 114) = 0.45, *p* > *0*.05).

Figure 2 displays the mean of the mediolateral body sway (axis X) and mean of the anteroposterior body sway (axis Y) in pretreatment, post-treatment, and follow-up for both intervention groups (rotational and vertical WBV) and for the control group.

Regarding the dynamic balance, no significant differences were found in any of the measured parameters, except gait speed, where the ANOVA revealed significant effects due to TIME (2, 114) = 12.4, *p* < 0.001) and GROUP x TIME (F(4, 114) = 4.41, *p* < 0.05). Post hoc comparisons revealed that participants in the rotational WBV group significantly increased their scores from the pre- to the post-treatment sessions (*p* < 0.001), but these scores decreased from the post-treatment to the follow-up (*p* < 0.001). The rotational WBV group was faster than the vertical WBV and the control group in the post-treatment (*p* < 0.05) but not in the follow-up (*p* > 0.05). There were no significant effects due to GROUP (*p* > 0.05).

## 4. Discussion

The present study has compared the effectiveness of two types of WBV, one vertical and one rotational, through a 12-week training in patients with fibromyalgia, in comparison with a control group that did not perform any training. This comparison was carried out on various fibromyalgia outcomes: self-report questionnaires on fibromyalgia impact [29], pain [30] quality of life [31], vibration sensitivity [34], functional motor capacity [35,36,37,38] (Berg Scale, 6-min walking test, and isometric back muscle strength), and balance [35]. The therapy program based on both WBV platforms showed differences in terms of results. The use of the rotational WBV showed greater effectiveness, reporting improvements in greater number of parameters than the vertical WBV. In both cases, these benefits were not maintained three months after the end of the program. To our knowledge, this is the first study that has compared the effects of two types of WBV platforms without combination with an exercise program and that has added a follow-up to assess whether the effect achieved is maintained over time.

Our findings are like those reported previously that have indicated that both types of WBV combined with an exercise program reduced pain and increased quality of life in patients with fibromyalgia [15,16,22,26,42] or osteoarthritis [6,7,8,9]. Regarding the perception of pain, measured with the analogue visual scale and in somatosensory measures such as pain thresholds, there were significant differences in both types of WBV with respect to the control group. These improvements in pain perception are in line with other studies [35,43]. Vibration strongly affects the afferent discharge of the quick adaptation mechanoreceptors and the muscle spindles. The activation of A-alpha fibers produced during vibration may compete with the central and peripheral nociceptive activity in the dorsal horn of the spinal cord, resulting in a reduction of second-order nociceptive activity in line with the Gait Control Theory [44], with the consequent decrease in the perception of pain [45,46]. In addition, both types of vibratory platforms produce a “tonic vibratory reflex”, which increases motor unit recruitment and induces more efficient use of the positive proprioceptive feedback loop, leading to better postural control and decreased pain [11]. This improvement in pain leads to an improvement in the quality of life in these patients, also observed in other studies [14,15,21,26,38].

Concerning vibration thresholds in the index fingers and toes, there was only a significant improvement after the use of the rotational WBV and not with the vertical one. The lack of absorption of vibrational energy with the vertical WBV, even when the vibration frequency has been limited for therapeutic use, may be the cause of this disadvantage. In addition, it has been observed that secondary terminations are less sensitive to vibration and respond effectively when less mechanical load reaches them [47], as occurs in the rotational WBV.

In motor function tasks, mediolateral sway, and gait speed, there were significant improvements only in the post-treatment of the rotational WBV group, while in the vertical WBV group and control group, no significant changes were observed. These facts are consistent with other studies that used rotational WBV [21,22,48] and can be explained by several facts: rotational WBV movement mimics the natural rotation of the hips during gait, which can activate a greater variety of muscles using physiological movement patterns [17] and generates more muscle stimulation compared to vertical WBV [49,50]. The additional instability of the lateral alternating vibrations of rotational WBV poses a more challenging training condition for postural control of mediolateral sway [51]. In addition, it has been observed that the neuromuscular activity measured by electromyography improves during rotational WBV compared to vertical WBV or a control group, recruiting more motor units [52,53]. This greater muscle activation generated by rotational WBV can result in increased isometric activity of the musculature [54]. Likewise, it has been shown that accelerations in the ankle are greater during rotational WBV [55,56]. This greater mechanical load on the ankle produced with rotational WBV can further activate the musculature at this level, especially the anterior tibial, whose neural excitability is highly associated with mediolateral oscillation [57] and gastrocnemius [53,58]. This musculature is considered essential in maintaining posture [59] and can improve joint stabilization and proprioception [16,18,60].

It is important to report that, after using the vertical WBV in our study, three patients had negative side effects. Thus, two patients had a momentary reduced capacity for visual-motor monitoring (difficulty in following moving objects) after the use of the vertical WBV, and one patient experienced reduced visual acuity for a moment (her vision became blurred). This is critical to consider, since the studies carried out have not made significant calculations of the harmful effects of this therapy, and as such definitive conclusions on the safety of WBV have not been obtained [20,25]. An important limitation of vertical WBV is that it moves the entire body up and down, including the head. On the rotational platform, the pelvis tilts and the other leg falls, keeping head movement to a minimum. However, most of the vibrational energy produced by the vertical WBV is not absorbed when it reaches the head, so that it causes much more head acceleration with respect to rotational WBV [18] and can cause negative side effects.

Some limitations must be considered for the adequate interpretation of the results. Although medication was controlled, it was not suppressed in participants with fibromyalgia, and opioids, tricyclics, or benzodiazepines have been demonstrated to have side effects on the outcomes we measured. Likewise, further research is necessary to know if these improvements occur at the beginning of the first sessions or a minimum amount is required

## 5. Conclusions

The comparison between rotational and vertical WBV revealed differences in the effects of the mechanical induction of vibration stimuli to the human body. The use of the two types of WBV reported benefits over the control group in most outcomes. However, in the vibration thresholds, motor function tasks, mediolateral sway, and gait speed, there was only a significant improvement after the use of the rotational WBV and not with the vertical one. Nevertheless, positive results seemed not to last for either of the two modalities, which makes it necessary to continue investigating until a protocol is found that has a permanent impact on the central nervous system. Meanwhile, the use of rotational WBV continuously or intermittently can be recommended.

## Figures and Tables

**Figure 1 ijerph-18-03007-f001:**
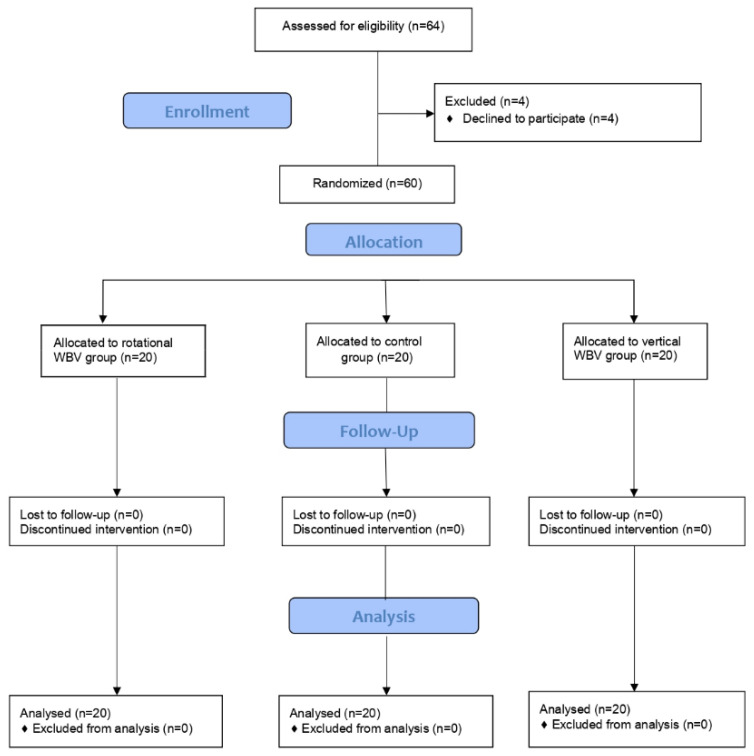
CONSORT flow diagram.

**Figure 2 ijerph-18-03007-f002:**
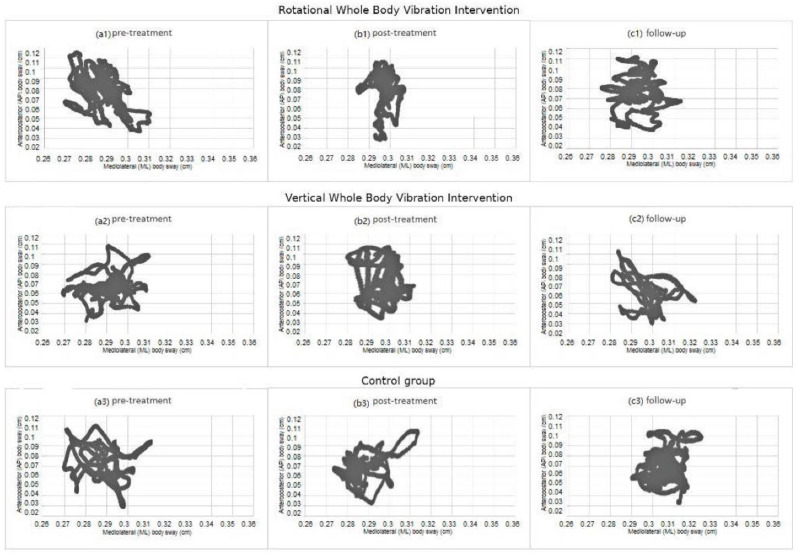
Mean of mediolateral body sway (axis X) and mean of anteroposterior body sway (axis Y) in pretreatment, post-treatment, and follow-up for the intervention groups and for the control group.

**Table 1 ijerph-18-03007-t001:** Sociodemographic data from the intervention and control groups. Sd: standard deviation, BMI: body mass index.

	CG (*n* = 20)	RWBV (*n* = 20)	VWBV (*n* = 20)	
	Mean ± sd	Mean ± sd	Mean ± sd	*p*
Age (years)	50.25 ± 8.53	52.30 ± 8.04	54.85 ± 8.62	0.23
BMI	23.34 ± 1.23	22.95 ± 1.30	24.21 ± 3.93	0.27
Height (centimeters)	169.15 ± 6.41	168.25 ± 6.35	166.90 ± 7.86	0.58
Weight (kilograms)	67.00 ± 7.46	65.05 ± 5.82	67.00 ± 7.43	0.59
Pain duration (years)	7.50 ± 3.22	6.75 ± 2.29	7.90 ± 2.82	0.42

CG: control group; RWBV: rotational whole body vibration; VWBV: vertical whole body vibration.

**Table 2 ijerph-18-03007-t002:** Comparisons results.

Dependent Variables	Vertical Group (*n* = 20)			Rotational Group (*n* = 20)			Control Group (*n* = 20)		
	Baseline	After	Follow-up	Baseline	After	Follow-up	Baseline	After	Follow-up
Fibromyalgia Impact Questionnaire	81.07 (77.74–84.39)	77.68 (74.36–81.00)	80.73 (77.22–84.24)	81.87 (75.84–87.91)	69.37 (62.24–76.50)	81.00 (74.70–87.32)	81.44 (78.30–84.58)	81.72 (78.61–84.82)	81.67 (78.63–84.71)
Visual Analogue Pain Scale	7.72 (7.48–7.97)	7.47 (7.19–7.76)	7.77 (7.54–8.00)	7.75 (7.54–7.96)	7.12 (6.75–7.49)	7.81 (7.55–8.07)	7.80 (7.55–8.04)	7.80 (7.55–8.04)	7.78 (7.56–8.01)
Quality of Life Index	3.80 (3.32–4.27)	4.20 (3.81–4.59)	3.85 (3.39–4.31)	3.75 (3.35–4.15)	5.00 (4.54–5.45)	3.80 (3.33–4.27)	3.85 (3.34–4.36)	3.80 (3.33–4.27)	3.75 (3.25–4.25)
Pressure pain sensitivity epicondyles	21.45 (16.56–26.35)	22.95 (17.99–27.92)	21.65 (16.89–26.42)	21.07 (17.16–24.98)	29.81 (25.24–34.38)	21.49 (17.65–25.32)	21.90 (18.77–25.03)	22.19 (19.25–25.13)	22.04 (18.83–25.25)
Pressure pain sensitivity index fingers	36.08 (28.92–43.24)	38.36 (30.98–45.75)	35.77 (28.51–43.02)	35.33 (30.49–40.16)	47.21 (43.19–51.22)	35.81 (31.33–40.30)	35.45 (29.24–41.66)	35.32 (28.95–41.68)	35.49 (29.25–41.73)
Vibration thresholds index fingers	3.61 (3.11–4.12)	3.40 (2.92–3.88)	3.64 (3.14–4.13)	3.52 (3.11–3.93)	2.84 (2.48–3.21)	3.57 (3.20–3.93)	3.57 (3.32–3.82)	3.52 (3.22–3.82)	3.55 (3.26–3.85)
Vibration thresholds toes	4.40 (3.89–4.90)	4.28 (3.78–4.78)	4.37 (3.86–4.87)	4.37 (3.94–4.81)	3.68 (3.30–4.07)	4.43 (3.88–4.98)	4.51 (4.27–4.74)	4.50 (4.24–4.76)	4.50 (4.25–4.75)
Berg Scale	27.95 (25.79–30.11)	29.60 (27.41–31.79)	28.15 (26.11–30.19)	27.00 (24.27–29.73)	39.10 (36.81–41.39)	27.25 (24.52–29.98)	28.15 (26.68–29.62)	28.45 (27.01–29.89)	28.40 (26.94–29.86)
Six-minute walking test	385.00 (353.15–416.85)	394.25 (362.55–425.95)	382.25 (349.86–414.64)	365.00 (354.00–376.00)	415.00 (402.76–427.24)	358.75 (342.95–374.55)	391.25 (369.35–413.15)	387.75 (367.70–407.80)	383.75 (364.79–402.71)
Isometric back muscle strength	33.85 (31.30–36.40)	35.70 (33.23–38.17)	32.75 (30.04–35.46)	33.40 (31.57–35.23)	40.85 (38.07–43.63)	33.45 (31.88–35.02)	33.05 (30.64–35.46)	32.85 (30.57–35.13)	33.10 (29.85–36.35)
Mean sway velocity	0.019 (0.014–0.023)	0.013 (0.008–0.018)	0.018 (0.013–0.023)	0.017 (0.013–0.021)	0.011 (0.008–0.013)	0.016 (0.013–0.019)	0.018 (0.013–0.021)	0.017 (0.013–0.021)	0.017 (0.015–0.020)
Mediolateral body sway	0.016 (0.012–0.019)	0.011 (0.007–0.015)	0.014 (0.010–0.019)	0.013 (0.008–0.018)	0.004 (0.002–0.005)	0.013 (0.006–0.021)	0.014 (0.009–0.019)	0.013 (0.008–0.017)	0.012 (0.008–0.016)
Anteroposterior body sway	0.013 (0.011–0.015)	0.012 (0.009–0.014)	0.013 (0.009–0.016)	0.015 (0.012–0.017)	0.013 (0.009–0.016)	0.015 (0.011–0.019)	0.013 (0.009–0.016)	0.012 (0.008–0.016)	0.012 (0.008–0.016)
Gait speed	3.01 (2.57–3.45)	3.48 (3.05–3.91)	3.02 (2.53–3.51)	2.99 (2.69–3.30)	3.93 (3.40–4.44)	3.19 (2.80–3.58)	3.04 (2.58–3.49)	3.02 (2.58–3.46)	3.06 (2.55–3.57)
Stride length	0.97 (0.83–1.13)	1.00 (0.85–1.15)	0.93 (0.80–1.06)	0.88 (0.72–1.03)	0.95 (0.82–1.08)	0.93 (0.77–1.08)	0.93 (0.77–1.10)	1.13 (0.94–1.31)	1.05 (0.88–1.22)
Percentage of time in the stance phase	68.30 (65.53–71.07)	67.41 (65.44–69.37)	68.76 (66.35–71.16)	64.75 (57.51–71.99)	64.84 (60.84–68.85)	67.56 (65.64–69.47)	68.48 (65.04–71.91)	67.56 (63.05–72.07)	67.94 (64.59–71.29)
Percentage of time in the swing phase	31.69 (28.92–34.47)	32.59 (30.62–34.55)	31.24 (28.84–33.64)	31.77 (29.78–33.76)	33.45 (31.68–35.22)	32.44 (30.52–34.35)	31.52 (28.08–34.95)	33.93 (30.96–36.90)	32.25 (28.85–35.65)

## Data Availability

The data presented in this study are available on request from the corresponding author.

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
