# Peer review of "A Comparison of the Effect of Two Types of Whole Body Vibration Platforms on Fibromyalgia. A Randomized Controlled Trial"

_ijerph, 2021, doi:10.3390/ijerph18063007_

Round 1

Reviewer 1 Report

I want to congratulate the authors for a very interesting manuscript that I consider necessary for the topic. However, an extensive revision of the manuscript is required previous to consider its publication:

-Authors should pay attention to the template: in the first page, in the citation paragraph at the left, the name of the Journal "Sensors" appear instead of "IJERPH". Check also the page headings.

-Abstract begins by the Methods section, avoiding any mention to the background of justification of the study. Please consider to include information that justify the use of whole body vibration in fibromyalgia.

-Introduction give enough information about fibromyalgia and whole body vibration systems. However, a connection between them is needed. Please add information explaining why whole body vibration could influence in pain or in the symptoms of fibromialgia patients. Referring that whole body vibration is useful in fibromyalgia is not a justification without explaining the evidences or hypothetical mechanisms involved.

Page 3 line 101: why did you used the criteria of 1990 when there are more recent criteria (2010, 2011, 2016)? In this line, I suggest to include some study referring that the criteria of 1990 correlate with the more recent ones.

Table 1: pay attention that according to the journal guide for authors, title of the tables must appear before the table and not after. Applicable to all the tables. It is also recommended to use acronyms at the table naming each group instead of long titles (if acronyms are used, they must be explained at footnotes).

Line 78-79: the objective of the study must be clear, please replace "novelty" by "objective".

The use of the CONSORT checklist is recommended to authors, as this guide is the internationally accepted for the report of clinical trials. Its used would have saved them from many of the comments of this revision.

Line 126: include the country in the model of platform, for example "Galileo model (Brand, city, state (if USA), country)". The same in line 158.

Lines 135, 138, 141: replace "3" by "three", as numbers under 10 should be written in letters.

Line 137: check an extra space between "leg." and "The".

Line 139: it must be explained who performed the treatments and the assessments to asure the masking process or blinding.

Lines 144-209: reliability and validity of the questionnaires and instruments must be mentioned, with confidence intervals when appropriate, as well as minor detectable change if known.

Lines 210-212: confidence interval used must be mentioned, as well as statistic assessment to asure the normal distribution of the groups.

Line 149: isn't it a 10cm line?

270: Check a double space

A table showing the results of each group would be of help for any reader. Anyway, this is just an advice for authors.

316-320: please include the reference of the variables measured.

Line 334: "A fibers" or "alpha fibers"? Please check.

Line 364: I want to congratulate the authors for this important paragraph. Adverse side effects are not always considered.

Line 375: a limitations paragraph is needed, or do authors consider that their study is perfect and can not be improved? 

Line 377-378: conclusion should be more concise, informing in what variables positive effects were observed for each whole body vibration, according to the objective proposed in line 78.

Line 381-382: this is not a conclusion, this must be placed in the last paragraph of discussion (limitations and future research).

Line 383: please include a space line after conclusions.

Author Response

I want to congratulate the authors for a very interesting manuscript that I consider necessary for the topic. However, an extensive revision of the manuscript is required previous to consider its publication:

-Authors should pay attention to the template: in the first page, in the citation paragraph at the left, the name of the Journal "Sensors" appear instead of "IJERPH". Check also the page headings.

“Sensors" has been changed to IJERPH in the citation paragraph and in the headings.

-Abstract begins by the Methods section, avoiding any mention to the background of justification of the study. Please consider to include information that justify the use of whole body vibration in fibromyalgia.

The rationale for the use of whole body vibration in fibromyalgia patients has been included in the abstract.

-Introduction give enough information about fibromyalgia and whole body vibration systems. However, a connection between them is needed. Please add information explaining why whole body vibration could influence in pain or in the symptoms of fibromialgia patients. Referring that whole body vibration is useful in fibromyalgia is not a justification without explaining the evidences or hypothetical mechanisms involved.

An explanation of the mechanisms involved has been added.

Page 3 line 101: why did you used the criteria of 1990 when there are more recent criteria (2010, 2011, 2016)? In this line, I suggest to include some study referring that the criteria of 1990 correlate with the more recent ones.

This was a mistake. The criteria of the American College of Rheumatology 2016 were used. It has been corrected in the text and in the bibliography.

Table 1: pay attention that according to the journal guide for authors, title of the tables must appear before the table and not after. Applicable to all the tables. It is also recommended to use acronyms at the table naming each group instead of long titles (if acronyms are used, they must be explained at footnotes).

The title has been placed before Table 1. Acronyms have been used in the table and explained in footnotes.

Line 78-79: the objective of the study must be clear, please replace "novelty" by "objective".

"Novelty" has been replaced by "objective".

The use of the CONSORT checklist is recommended to authors, as this guide is the internationally accepted for the report of clinical trials. Its used would have saved them from many of the comments of this revision.

The article has been revised following the CONSORT checklist.

Line 126: include the country in the model of platform, for example "Galileo model (Brand, city, state (if USA), country)". The same in line 158.

Countries have been included in both models.

Lines 135, 138, 141: replace "3" by "three", as numbers under 10 should be written in letters.

"3" has been replaced by "three"

Line 137: check an extra space between "leg." and "The".

The extra space has been checked.

Line 139: it must be explained who performed the treatments and the assessments to asure the masking process or blinding.

It has been explained who performed the treatment and the assessments.

Lines 144-209: reliability and validity of the questionnaires and instruments must be mentioned, with confidence intervals when appropriate, as well as minor detectable change if known.

The reliability and validity of the questionnaires and instruments has been mentioned.

Lines 210-212: confidence interval used must be mentioned, as well as statistic assessment to asure the normal distribution of the groups.

The confidence interval used has been mentioned, as well as the statistical evaluation to ensure the normal distribution of the groups.

Line 149: isn't it a 10cm line?

This was a mistake. It has been corrected.

270: Check a double space

The extra space has been checked.

316-320: please include the reference of the variables measured.

The references of the variables have been included.

Line 334: "A fibers" or "alpha fibers"? Please check.

“A fibers” has been replaced by “A-alpha fibers”.

Line 364: I want to congratulate the authors for this important paragraph. Adverse side effects are not always considered.

Thanks for your appreciation.

Line 375: a limitations paragraph is needed, or do authors consider that their study is perfect and can not be improved? 

A limitations paragraph has been added.

Line 377-378: conclusion should be more concise, informing in what variables positive effects were observed for each whole body vibration, according to the objective proposed in line 78.

A concise conclusion has been added.

Line 381-382: this is not a conclusion, this must be placed in the last paragraph of discussion (limitations and future research).

This sentence has been placed in the last paragraph of discussion.

Line 383: please include a space line after conclusions.

A space line after conclusions has been added.

Reviewer 2 Report

abstract
Start with the goal. The type of study must be moved to the beginning of the method. Include participants' gender.
Include the assessment tools for the dependent variables.
Put the p value whenever there are statistical differences
Do not repeat keywords that are already in the title to improve the article search

Introduction
It is important to refer to epidemiology in relation to sex.
Reference the justifications by which WBV improves pain and function in patients with AO and rheumatoid attrition.
Explain how and why, based on physiology / pathology / biomechanics, WBV can help patients with fibromyalgia. This is essential to justify the need for the study.

Methods
The 1st paragraph of the method must be relocated in the introduction section
Include the sample calculation formula used
Who made the diagnosis and selection of volunteers?
How was randomization done?
Table 1 should be reallocated for the results and in the method just say that the groups were matched
Were the platforms calibrated to ascertain their frequency and functionality as proposed?
Was the exam room air conditioned?
Has the schedule for both experimental groups been standardized? It is known that the schedule of physical activity impacts performance.
Who performed or instructed the intervention? was trained for ALL conditions and analyzes?
Was the level of pre-intervention physical activity verified? Did anyone perform physical therapy treatment before the intervention?
Better describe the assessment tools. This is very summarized.
The description of the statistical analysis is very poor. It must contain: program used, analyzes performed, p value considered significant. It is suggested to perform other tests such as linear regression.

Results
Present the results of the comparisons all in tables.
The result of the regression in a separate table.

Discussion
Good, it is suggested to further explore the justifications of movement neuroscience to improve pain and QOL, as well as studies that indicate exercises to improve AO.

Author Response

ABSTRACT

Start with the goal.

It has started with the goal.

The type of study must be moved to the beginning of the method.

The type of study has been moved to the beginning of methods.

Include participants' gender.

Participants’ gender has been included.

Include the assessment tools for the dependent variables.

These tools have been included.

Put the p value whenever there are statistical differences.

Added p-value for statistical differences.

Do not repeat keywords that are already in the title to improve the article search.

Fibromyalgia has been removed as a keyword.

INTRODUCTION
It is important to refer to epidemiology in relation to sex.

Epidemiological reference has been added in relation to sex.

Reference the justifications by which WBV improves pain and function in patients with AO and rheumatoid attrition.

Reference has been made to this point.

Explain how and why, based on physiology / pathology / biomechanics, WBV can help patients with fibromyalgia. This is essential to justify the need for the study.

This point has been explained.

METHODS
The 1st paragraph of the method must be relocated in the introduction section.

It has been recolocated.

Include the sample calculation formula used.

The sample calculation formula has been added.

Who made the diagnosis and selection of volunteers?

It has been added that the volunteers were diagnosed by doctors and selected in the Associations by an external researcher.

How was randomization done?

Randomization explanation added.

Table 1 should be reallocated for the results and in the method just say that the groups were matched

The Table 1 has been reallocated for the results and it has been mentioned in method that the three groups were matched.

Were the platforms calibrated to ascertain their frequency and functionality as proposed?

It has been specified that the two platforms were calibrated to determine their frequency and functionality.

Was the exam room air conditioned?

It has been specified that the therapeutic program was performed in an air conditioned room.

Has the schedule for both experimental groups been standardized? It is known that the schedule of physical activity impacts performance.

It has been specified that it was a standardized program that was carried out at the same time in the afternoon.

Who performed or instructed the intervention? was trained for ALL conditions and analyzes?

It has been specified that it was performed by a trained external physiotherapist for all conditions of the intervention.

Was the level of pre-intervention physical activity verified? Did anyone perform physical therapy treatment before the intervention?

It has been specified that none of the volunteers underwent physiotherapy treatment or physical exercise before the intervention. The level of previous functionality was verified by functional tests such as the Berg scale, the six-minute walk test and the isometric strength of the back muscles.

Better describe the assessment tools. This is very summarized.

Assessment tools have been better described.

The description of the statistical analysis is very poor. It must contain program used, analyzes performed, p value considered significant. It is suggested to perform other tests such as linear regression.

The statistical analysis description has been expanded.

RESULTS
Present the results of the comparisons all in tables.
The result of the regression in a separate table.

All the results of the comparisons have been presented in Table 2.

DISCUSSION
Good, it is suggested to further explore the justifications of movement neuroscience to improve pain and QOL, as well as studies that indicate exercises to improve AO.

Justifications for pain improvement and quality of life have been added.

Round 2

Reviewer 1 Report

I want to congratulate the authors for their work at implementing all the suggestions.